# Farmers Perceptions of Climate Change Related Events in Shendam and Riyom, Nigeria

## Simi Goyol [1,*] and Chaminda Pathirage [2]

[1] Centre for Disaster Resilience, School of the Built Environment, University of Salford, Salford M5 4WT, UK
[2] School of Architecture and Built Environment, University of Wolverhampton,
Wolverhampton WV1 1LY, UK; C.Pathirage@wlv.ac.uk
* Correspondence: s.goyol@edu.salford.ac.uk

**Abstract:** Although agriculture in Nigeria is the major source of income for about 70% of the active population, the impact of agrarian infrastructure on boosting productivity and supporting livelihoods has increased. Climate change and the increasing trend of climate-related events in Nigeria challenge both the stability of agrarian infrastructure and livelihood systems. Based on case studies of two local communities in Plateau state in Nigeria, this paper utilizes a range of perceptions to examine the impacts of climate-related events on agrarian infrastructures and how agrarian livelihood systems are, in turn, affected. Data are obtained from a questionnaire survey (*n* = 175 farmers) and semi-structured interviews (*n* = 14 key informants). The study identifies local indicators of climate change, high risks climate events and the components of agrarian infrastructures that are at risk from climate events. Findings reveal that, changes in rainfall and temperature patterns increase the probability of floods and droughts. They also reveal that, although locational differences account for the high impact of floods on road transport systems and droughts on irrigation infrastructures, both have a chain of negative effects on agricultural activities, economic activities and livelihood systems. A binomial logistic regression model is used to predict the perceived impact levels of floods and droughts, while an in-depth analysis is utilized to corroborate the quantitative results. The paper further stresses the need to strengthen the institutional capacity for risk reduction through the provision of resilient infrastructures, as the poor conditions of agrarian infrastructure were identified as dominant factors on the high impact levels.

**Keywords:** agrarian infrastructure (AI); agrarian livelihoods; climate change; cascading effects; perception

**JEL Classification:** Q54

## 1. Introduction

Climate change, a major driver of natural hazards, is a global threat to human and economic development. Climate change is a shift from the average weather conditions over a period of time, which leads to unpredictable patterns and an increased occurrence of climate-related events. Current climate changes reveal rising temperatures, higher evaporation rates, and altered rainfall patterns (Cooper et al. 2008; Settele et al. 2015). Whilst this accounts for heavier rains, surface runoff and consequent floods, low rainfall, and water shortages lead to drier conditions and droughts. Projections suggest that the persistent alteration of the climate system is likely to be prolonged due to increasing variations in the average weather conditions which will increase the occurrences of climate-related events, such as floods and droughts (Dai and Zhao 2017). This is expected to further challenge global economies; studies in various sectors, including transport (Nemry and Demirel 2012; Neumann et al. 2015), power

(Panteli and Mancarella 2015; Van Vliet et al. 2016), water (Olmstead 2014; Olmstead et al. 2016; Van Vliet et al. 2013) and agricultural (Ghile et al. 2014; Kurukulasuriya and Rosenthal 2013) have documented evidence of climate change impacts.

Agriculture plays a fundamental role in providing food for growing populations, raw materials for industries and support to agrarian livelihood systems (Hertel and Lobell 2014). This, in turn, contributes to the growth of a country's GDP, sustains economic development and significantly reduces poverty levels (Binswanger and Landell-Mills 2016; Godoy and Dewbre 2010). Human population growth, accompanied by the need for economic support, demands a rise in agricultural production to meet increasing demands (Gerland et al. 2014). Agrarian infrastructure, including physical assets and service systems, are vital for improved agricultural production and the smooth running of agrarian communities. Unfortunately, such infrastructures in Nigeria are scarce, and the few available are in a poor condition and are prone to damage by climate change-related events (Ayinde et al. 2010; Ebele and Emodi 2016). How future climate change will affect agrarian infrastructures is uncertain. Kurukulasuriya and Rosenthal (2013) argued that the potential impacts of climate change would have implications for the agricultural sector by affecting agricultural production and particularly agrarian infrastructure systems. This will not only undermine the performance of the sector, but also unleash future risks and uncertainty regarding the function of agrarian infrastructure systems.

Furthermore, infrastructure sectors, including the agriculture sector, are interdependent for their functioning, so that damage to an individual element can precipitate disruption within the system (Chappin and van der Lei 2014). Regardless of the nature of infrastructure interdependencies, a chain of negative events, also referred to as a "cascading effect", can be initiated (Pescaroli and Alexander 2016). Hence, damage to the infrastructure system by adverse climate change will have implications for individual elements and on a wider scale, thereby affecting human and economic development efforts. This highlights the urgent need to reduce the risk from uncertainties due to climate change; therefore, the aim of this paper is to examine the impacts of climate-related events on agrarian infrastructure and its cascading effect on agrarian livelihood systems. This aim is achieved by the following objectives: Firstly, by identifying the local indicators of climate change and their relationship to climate-related events. Secondly, by assessing the impacts of climate-related events on agrarian infrastructure and the cascading effect of infrastructure disruption on agrarian livelihood systems. Thirdly, by analyzing the factors influencing perceived levels of climate impact. This paper therefore contributes to the understanding of how perceptions can enhance adaptation and sustain agrarian livelihood systems.

## 2. Agrarian Infrastructure

Infrastructure generally refers to the basic physical facilities and organizational structures necessary for a society or economy to function effectively. Agricultural infrastructure, often used interchangeably with agrarian infrastructure (AI), manifests either as hard physical facilities, such as transportation networks, or as soft service systems. AI is crucial for continuous food production and the sustainability of the rural economy. Several studies, such as those by Antle (1983), Binswanger et al. (1993), Pinstrup-Andersen and Shimokawa (2008), Venkatachalam (2003) and Zhang and Fan (2004), concluded that the availability of AI in rural areas has a clear influence on agricultural production, including a sustainable supply chain of agricultural goods and other non-farm activities. Shenggen and Zhang (2004) reports that infrastructure investment is a major determinant of economic development and particularly of growth in the agricultural sector. Wharton (1967) and Patel (2014) provided various classifications in order to gain insights into the context of AI for increased agricultural productivity (Table 1). Wharton's classification includes capital intensive, capital extensive and institutional infrastructure, while Patel classified these as input based, resource based, and physical and institutional infrastructures.

**Table 1.** Classification of Agricultural Infrastructure (Extracted from: Wharton 1967; Patel 2014).

| Wharton (1967) | Patel (2014) |
|---|---|
| Capital Intensive: Irrigation, Roads, Bridges | Physical Infrastructure: Road connectivity, Transport, Storage, Processing, Preservation. Resource based: Water/Irrigation, Farm power/Energy |
| Capital Extensive: Extension Services | Input based: Seed, Fertilizer, Pesticides, Farm equipment, and Machinery. |
| Institutional: Formal & Informal institutions | Institutional Infrastructure: Agriculture research, Extension & Education Technology, Information & communication services, financial services, marketing |

Both Wharton and Patel recognized two critical roles: firstly, physical infrastructure, such as roads and irrigation facilities in determining the extent of agricultural output, and secondly, institutions, such as agricultural extensions, research and input services, which provide services to improve crop yields by enhancing the application of scientific knowledge. Building on these emphases, this paper narrows agriculture to crop production, and recognizes that AI can be classified according to domains. These domains are broadly categorized into on-farm and off-farm infrastructures, and include:

- Off-Farm infrastructure:

    - Transport systems (roads and bridges)
    - Institutional service systems (agricultural research and extension services)

- On-Farm Infrastructure:

    - Irrigation systems (dams, tube wells, boreholes)
    - Inputs (fertilizer, seeds, and farm implements)

Both off-farm and on-farm AI significantly boosts levels of production and subsequently stimulates the rural economy. While off-farm facilities and services may not be located at the point of production, they influence the input-production-output links. Gajigo and Lukoma (2011) emphasized the importance of links, such as access to inputs, improving outputs, reducing transaction costs, and connecting global markets. Similarly, Townsend (2015) asserts that these links improve agriculture, which in turn reduces 65% of rural poverty, improves food security, and raises income levels. A farm infrastructure includes assets and facilities that enhance agricultural intensification to improve productivity. In five case studies, Maraseni et al. (2012) demonstrated the ways in which on-farm infrastructure positively influences savings on labor and water use, increases productivity, and encourages a good return on investment. Similarly, Garnett et al. (2013) and Diao (2016) opined that these have a high correlation with economic growth.

Similar studies conducted in Nigeria by Adesugba and Mavrotas (2016), Sertoglu et al. (2017) and the National Bureau for Statistics Nigeria (2017) stated that the Nigerian agricultural sector plays a vital role in economic development by significantly contributing to the nation's GDP. For over 30 years (1980–2012), the sector consistently contributed above 30% to the GDP (Ahungwa et al. 2014). However, in 2016, although the sector remained a major employer of about 70% of the active population, its contribution to the nation's GDP declined to 23% (Federal Ministry of Agriculture and Rural Development 2016). Literature (Adegbenle and Olatunji 2017; Effiom and Ubi 2016; Goyol et al. 2017; Rufus and Bufumoh 2017) suggests that infrastructure decay is a major challenge to national development and influences the decline of the sector's contribution to the nation's GDP. Agriculture in Nigeria is highly dependent on rainfall, and is dominated by small hold farmers operating local farming methods at a subsistence level. In addition, the poor state of infrastructure that supports agricultural production, in the form of hard physical facilities and soft service systems, is a threat to the performance of the sector (Obadiah et al. 2016). These challenges, alongside shocks

from climate change, are hindrances to the development of the Nigerian agricultural sector. Future climate change, which is expected to raise the magnitude and severity of climate-related hazards, will have implications not only for crop yields, but also on AI for sustained productivity (Elliott et al. 2014; Sun et al. 2013).

*Agricultural Infrastructure Policies and Management in Nigeria*

Until recently, the general provision and management of infrastructure in Nigeria was the sole responsibility of the government, and was conducted through a vertical relationship between the three government tiers. However, in recent years, the government adjusted certain policies to open up opportunities for private partnership (Udoka 2013). Moreover, Adeyinka and Omotayo (2015) observed that this has not been fully implemented, as only about 15% accrues due to public private partnership. Although the Nigerian government recognizes the importance of infrastructure investment and has shown increasing interest in demonstrating a higher political commitment to invest, particularly in the agricultural sector, Daze et al. (2011) observed that, in reality, the details of plans are quite different from those presented in policy documents.

In Nigeria, the Federal Ministry of Agriculture and Rural Development (FMARD) has, over the years, provided strategic guidance, sourced funds, and overseen the implementation of set goals at state and local levels. These were guided by government policies and programs such as the River Basin Development Authorities (RBDAs) management of irrigation schemes, the Directorate for Food, Roads and Rural Infrastructure's (DFRRI) management of rural roads and water supplies, Agricultural Development Projects (ADPs), and the National Fadama Development Project (NFDP) which aims to improve rural development by fostering agricultural productivity and other non-farm activities. However, these programs suffered setbacks, as most were ineffective due to the mismanagement of resources and failed siting of projects, among other issues (Enplan Group 2004). For instance, the roads provided under such programs were poorly constructed, and water facilities were below capacity and could not last due to lack of maintenance (Fiki et al. 2007). In assessing the performance of these policies, Nchuchuwe and Adejuwon (2012) concluded that rural infrastructure provision still remains a concern despite several policy attempts. This failure is attributed to governments' activities, revealing that the priority in policy formulation and resource allocation is accorded to urban areas at the expense of rural areas.

Recently, policies such as the Agricultural Transformation Agenda 2011–2015 (ATA) and the Agricultural Promotion Policy 2016–2020 (APP) have aimed to improve agricultural development in a number of ways, including the improvement of infrastructure investments (Federal Ministry of Agriculture and Rural Development 2016). Under the ATA, FMARD works in collaboration with the state engage ministries whose responsibilities contribute to growth in agriculture and rural development. For instance, the State Ministry of Works provides technical services in design, construction and the maintenance of feeder roads. A World Bank report by Olomola et al. (2014) pointed out that the current limitation characterizing ATA policy is that only about 10% of the budget was allocated to the construction of feeder roads. Also, no consideration was made for the provision of other vital AIs, such as irrigation and storage facilities; this was partly because the administration for irrigation in Nigeria is not under the control of the FMARD (Federal Ministry of Agriculture and Rural Development 2016; Ifejika Speranza et al. 2018). The sectoral, rather than the integrated, approach to agrarian infrastructure management in Nigeria poses challenges to the realization of set policy objectives towards the growth of the sector. AI, such as feeder roads, waterways and irrigation facilities, rural electrification, storage facilities, and market facilities are generally given the lowest priority in infrastructure development. For instance, at present, about 70% of the 193,200 km of Nigeria's road network is in a deplorable state, which increases the cost of agricultural goods by between 50–100% (Federal Ministry of Agriculture and Rural Development 2016), and exceeds the 30–40% cost of trade goods in Africa (Gutman et al. 2015). Large dams and water reservoirs in Nigeria are constructed for combined purposes, although most are for water supply and hydro-electric power generation.

Irrigation farming in Nigeria depends on surface and sub-surface water sources from natural streams, ponds, wells, boreholes and small-scale motor pumps that irrigate crops. However, Takeshima (2016) observed that these small irrigation schemes and the small scale of production are major drivers for the low returns on investments, the high cost of labor and the high cost of market transactions. These challenges hinder efforts towards the realization of the set goals, leaving the agricultural sector in need of infrastructure facilities and service systems to support extensive production. The government recognizes these, and other, challenges, and intends to address them under the current APP.

## 3. Climate-Related Events and Their Impacts in Nigeria

Global warming is a driver of climate change and the increasing occurrences of climate-related events. Both natural and anthropogenic activities influence mean temperatures, evaporation and rainfall patterns, causing an imbalance in the earth's climate system. Several world regions are experiencing higher occurrences of adverse climate-related events, and developing regions, such as Africa, are particularly affected by these events (Alcamo et al. 2012; UNISDR 2004). Solomon (2007) observed that every 1 °C temperature rise will result in a 7% increase in evaporation and between 1–2% increase in precipitation. The literature (Allen 2015; Eruola et al. 2013; Gommes and Petrassi 1996; Salack et al. 2014) identified indicators of change in weather patterns which include warmer temperatures, higher evaporation rates, fluctuations in rainfall patterns, sporadic rains, shifts in the onset and cessation dates of rains, and extended dry periods. These and other similar patterns of hot-drier conditions are becoming common in Nigeria.

Nigeria, a sub-Saharan country, is located in the hot tropical climate zone, and bounded by the Sahara desert to the north and the Atlantic Ocean to the south. The country has two seasons (rainy and dry). Rainy periods range from between two to three months in the extreme north of the country and between 9–12 months in the coastal regions. These seasons largely determine the spatial variation in the mean maximum and minimum temperatures of 41–13 °C and 32–21 °C experienced in the north and south respectively (Adakayi and Ishaya 2016; Eludoyin et al. 2014). Moreover, less than 600 mm annual rainfall is recorded in the northern region (Akinsanola and Ogunjobi 2014). Rainfall is so far considered the most vital climate element for agriculture in Nigerian. This also determines the occurrences of either floods or drought events, both of which are a challenge to agriculture. Abiodun et al. (2013b) observed that rising temperatures, high evaporation rates and ocean currents account for the varying distribution of rainfall, which occurs less towards the extreme north and more along the coast of the country. These changes in the climate system, alongside other non-climate factors, are drivers of the occurrence of climate-related hazards, such as floods and droughts in Nigeria (Fuwape et al. 2016). A summary of the major climate-related events experienced in Nigeria shows that floods and droughts are particularly devastating in terms of the estimated cost of damage and the number of people affected (Table 2).

**Table 2.** Climate-related events in Nigeria 1900–2016.

| Event Type | Event Count | Total Deaths | Total Affected | Total Damage ($'$000 US$) |
|---|---|---|---|---|
| Droughts | 1 | 0 | 3,000,000 | 71,103 |
| Extreme Temperatures | 2 | 78 | - | - |
| Floods | 44 | 1493 | 10,478,919 | 644,522 |
| Storms | 6 | 254 | 17,012 | 2900 |

Source: The Emergency Events Database—Université Catholique de Louvain (UCL)—CRED, D. Guha-Sapir—www. emdat.be, Brussels, Belgium, (EM-DAT 2017).

The major climate-related hazards experienced in Nigeria are droughts, extreme temperatures, floods and storms. Due to the location of the country, there is a high variability in rainfall that often leads to either a deficit or an overflow. While drought is an identified period of low precipitation within an area leading to prolonged shortages of either atmospheric, surface or ground water, flooding

refers to the overflow of water into areas, which are usually dry. Droughts and floods are ranked highest in terms of the impact on human and economic activities. Although there are no records of extreme drought in the past few years, hotter and drier conditions are lowering water levels and shrinking water bodies leading to hydrological and agricultural droughts (Abiodun et al. 2013a; Olaniran and Sumner 1989, 1990). This agrees with the assertion by Ashton (2002) who projected that rising temperatures and high evaporation rates would heighten water demand for various purposes, including agricultural use. Also, Nyong et al. (2008); Rindap (2015) suggested that current water shortages and consequent droughts commonly experienced during the dry season do not only extend the fight against hunger but also trigger conflict in Nigeria. How climate change will affect the yield of water sources and the ability to meet water demand will have implications for irrigation agriculture since the sector is currently the largest consumer of water in Nigeria.

Floods are the most devastating climate-related event in terms of the magnitude and severity on human and economic activities. Adelekan and Asiyanbi (2016) reported that, within in 17 h, heavy rainfall of up to 233.3 mm in 2011 and 216.3 mm in 2012 lead to the Lagos floods in Nigeria. Frequent floods alongside rising sea level are threating coastal towns, major rivers are overflowing and even hinterlands, where such events were once rare, are now experiencing more floods. At least 20% of the population are at risk of flood, an average of 5000 persons are displaced by floods every year, whilst lives are lost annually. The past decade alone records that over 600 million US dollars were lost to floods in Nigeria (EM-DAT 2017). This affected various infrastructure sectors including transportation (Adelekan and Asiyanbi 2016). Road transportation is the main form of agricultural freight, moving food crops from agrarian communities to markets, processing points and urban centers in Nigeria; however this is increasingly at risk of flooding. Porter (2014) observed that unfortunately only about 30% of rural roads are all season roads, which affects the flow of both agricultural inputs and outputs. This confirms that agrarian roads are mostly unpaved feeder roads, characterized by laterite surfaces and poor drainage. Such poor conditions make them less resilient and vulnerable to damage by the increasing occurrence of flood events. Future projections suggest the increasing incidence of floods which will further threatens not only coastal towns but also hinterlands (Adeagbo et al. 2016; Davis 2013) will have implications for the poor state of agrarian infrastructure.

Recently, approaches to assessing climate change impacts have expanded from determining the risk of a climate event for an individual piece of infrastructure to researching an understanding of how vulnerabilities broaden the scope of impacts. Figure 1 illustrates the theoretical approach in assessing the impacts of climate-related events on agrarian infrastructure and the cascading effect on livelihood systems.

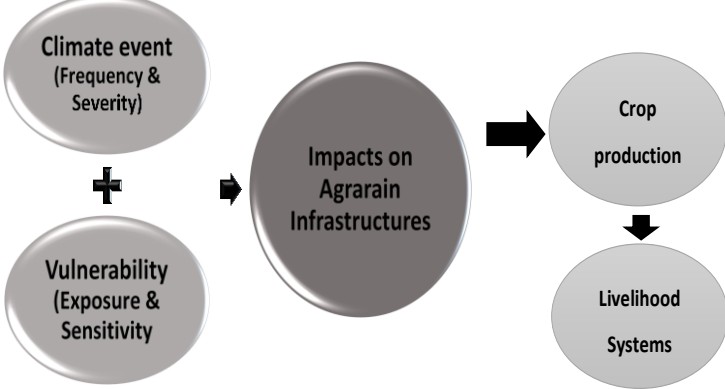

**Figure 1.** Theoretical Framework for Impact Assessment.

## 4. Methods

### 4.1. Study Area Description

Plateau state, located in North-Central Nigeria is an ecological transition zone that divides the semi-arid north from the forest south (Figure 2). The area has two distinct dry and rainy seasons. It experiences cool, semi-temperate weather with up to 1400 mm annual rainfall (Olaniran 2002), and a wide temperature range of between 8–25 °C (Odunuga and Badru 2015). Plateau varies geographically with highlands reaching 1200 m above the mean sea level and lowlands at approximately 200 m. Odumodu (1983) observed that this geographical variation is responsible for the wide variation of rainfall and temperature distribution, which means the area is at high risk of climate-related events. The state, similar to the surrounding Nigerian region, experiences changes in weather patterns and notable adverse events. Floods and droughts are two distinct climate-related hazards experienced in the state due to the wide variation in elevation, and its location within the ecological transition zone. These features are similar to those in other African regions (Brida et al. 2013; Devereux 2007; Van der Geest and Warner 2014). Floods, as a result of heavy rainfall and attendant surface run-off, are more frequent occurrences, particularly in the plateau lowlands. Research by Adewuyi and Olofin (2014) calculated that the central region alone, where Plateau state is located, recorded 31% of the 52 major flood incidences in Nigeria in 2012. Rural communities, where drainage systems are poor or absent, were worst affected. In terms of drought, although precipitation levels do not fall below the normal records, water shortages that lead to agricultural and hydrological droughts are evident (Tarhule 1997, 2007; Gongden and Lohdip 2009). These already have implications for irrigation farming in Plateau State.

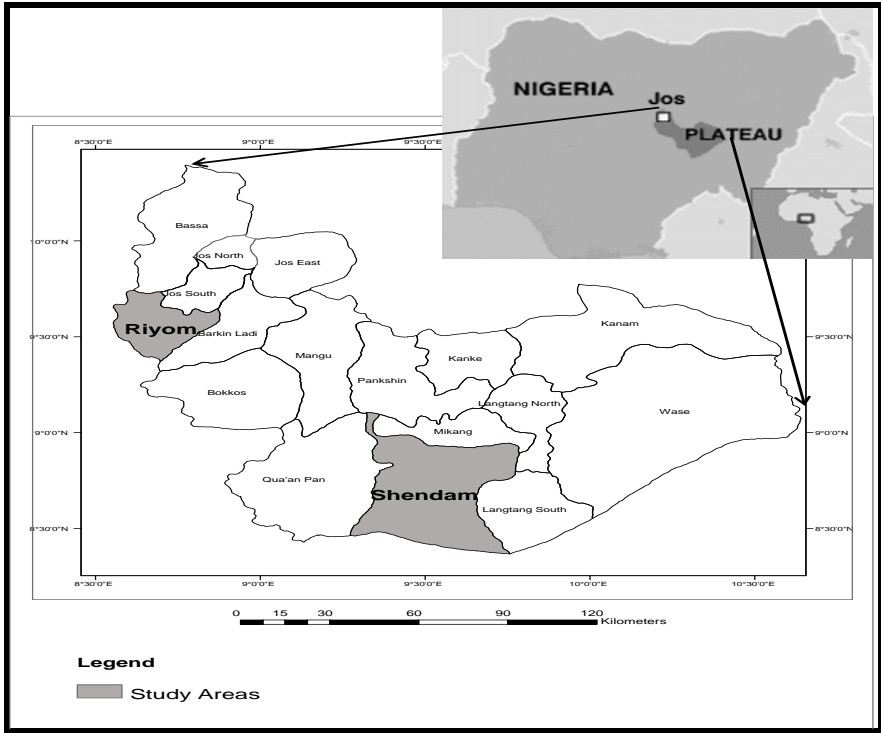

**Figure 2.** Study Area, Nigeria.

Alongside climate variations, Plateau State has a population of over 3.2 million (National Population Commission NPC), and 51% in rural areas engage in one form of agrarian activity. The area is endowed with both natural and man-made resources, and experiences weather which is conducive for the production of a variety of exotic fruits, vegetables and grain in the upland and tubers and rice in the lowlands. With the advantage of irrigation farming, these crops are grown

all year round, thereby contributing to the economic position of the country (Chuktu 2002). Although these are unique characteristics, Year-round crop cultivation is the bedrock for economic growth in agrarian communities, and as Tarhule (2007) states, they are the worst hit by drought and water scarcity. Audu et al. (2013) observed that weather shocks—both mean changes and extreme events—have negative implications for livelihoods and economic activities. Falaki et al. (2013), in analyzing climate change in the area, found that farmers' perceptions were based on their local experiences, and were corroborated by the climate records of changing temperatures and rainfall patterns which affects farm operations. Equally important, Dung-Gwom et al. (2008), Goyol and Pathirage (2017) and Wapwera (2014) noted that there is a physical dereliction of basic infrastructure, particularly in rural areas. There is a lack of infrastructure to support agricultural production, and that which is available is in a poor condition. The challenges associated with inadequate AI affect roads, irrigation facilities, agricultural extension and input services, storage and processing facilities. Furthermore, future climate change will increase demand, which will have further implications for weak infrastructures.

*4.2. Methodology*

According to the tenets of a pragmatic philosophical view point, this study adopted a concurrent mixed methods design that employed a combination of procedures to achieve a set aim (Johnson and Onwuegbuzie 2004; Onwuegbuzie et al. 2009; Teddlie and Tashakkori 2009). Both qualitative and quantitative data were used to supplement information sources and to ensure validity in this study. The first part of this research undertook a broad search on AI from which the on- and off-farm categories of infrastructure were identified, as they aim to increase production and ensure sustainable agrarian livelihoods. A search of the major climate-related hazards prominent in the region identified five major climate-related events, as indicated in Table 2. From this list, floods and drought were identified as the most significant devastating events which represent threats to AI assets, and as such, are selected for the study. From this, a comprehensive review of existing literature from journal articles, official documents, such as impact reports, and news publications, was undertaken to identify the infrastructure assets most affected by climate events and to inform the case study design. Thus, transportation systems, which are part of the off-farm infrastructure, and irrigation systems, which are part of the on-farm infrastructure, are recognized in the literature as being the most vulnerable to climate change.

Estimating global or regional climate changes to local scales can pose limitations to research findings. This is due to local differences in the topography, microclimate and other factors. In considering the future projections of climate change, this study adapts a climate event scenario approach to analyze the impacts of floods and droughts in two local communities, i.e., Shendam in case study 1 and Riyom in case study 2 (Figure 2). This selection was based on the extent of the impacts, the geographical location, the levels of infrastructural development and their accessibility. Although both communities are within agrarian areas, the differences in location account for different climate events, and hence, require a multi-hazard evaluation. In analyzing the impacts of floods and droughts on AI, the research adopted a cross-sectional study, which provided a view at a specific period of time. Because AI management is implemented through nationwide policies in a vertical manner, starting from the federal, state and local governments and then to community levels, a syllogism of institutional and community views were incorporated within the study.

4.2.1. Data Collection

A total of fourteen key semi-structured interviews were conducted with key informants using a purposive sampling technique. Twelve in-depth interviews were conducted with professionals responsible for AI management across the three tiers of government ministries and agencies. The interviewee's expertise and institutional records provided in-depth information on climate-related events and their impacts, as well as the institutional capacity for AI management. In addition, two interviews were conducted with community representatives who had firsthand knowledge of the

community and provided in-depth information on how climate-related impacts on AI affected agrarian livelihood systems. Furthermore, 175 farmers (69 in Shendam and 106 in Riyom) were surveyed as the main infrastructure users at a local level. The two selected case studies formed a natural strata; hence, a stratified random sampling technique with relative representation was employed to sample farmers' opinions through a face-to-face interview questionnaire. Information on the socio-economic characteristics of respondents was collected on a categorical scale, and questions on the local indicators of climate change, the impacts of climate-related events on AI, and the cascading effect on agrarian activities were designed on a Likert scale. Responses were grouped according to themes to quantify farmers' opinions on issues related to the local indicators of climate change, climate-related events, the impacts of climate events on agrarian transportation and irrigation infrastructure, the effects of AI failure on agricultural activities, and the cascading effect on agrarian livelihoods.

4.2.2. Data Analysis

Information from the transcribed interviews was subjected to a content analysis using NVivo software (version 11), where coded responses classified under themes were quantified based on the references and number of sources. After that, particular information from the two climate scenario events was extracted to corroborate the results of the quantitative analysis. An IBM SPSS 24 was used to analyze the survey questionnaire responses. Descriptive statistics of the results were presented in frequencies and percentages (suitable for categorical variables), and an inter-quartile range indicated the spread of the scale responses (Pallant 2016).

To establish the local indicators of climate change, a minimum of 10 years' farming experience was used to benchmark the time period for observed changes in local weather patterns. Where respondents had less than 10 years' farming experience, a matrix of 10 less their ages was used, as their knowledge of the area for over 10 years was found to be relevant. Percentage scores were then used to weight the indicators of climate change, and the mean scores were used to weight the perceptions of climate change impacts. High-risk indicators (90th percentile) were then analyzed on an impact scale and linked to past climate-related events in each case study location.

The binary logistic regression model was used to analyze likely factors influencing respondents' views on the impact (or no impact) of climate-related events on AI. In this case, the dichotomous variable was the farmer's opinion on the impact of a climate event or no impact, while the explanatory variables were the location/farming season $(x_i)$, sex $(x_{ii})$, age $(x_{iii})$, educational level $(x_{iv})$, farming years $(x_v)$, income level $(x_{vi})$, and percentage of farm income $(x_{vii})$.

## 5. Results and Discussion

### 5.1. The Description of Respondents Characteristics

The profiles of the fourteen key informants included their administrative level, professional background, gender and years of experience (Appendix A). The sample comprised the following: 2 individuals were at the federal level, 4 at the state: 4, 6 within local government, and 2 at the community level. Participants' professional backgrounds included technical, planning and mobilization/supervision at a 2:3:2 ratio respectively. The ratio of male to female participants was 5:1, with an average of 19 years work experience.

The socio-economic characteristics of farmers was collected from the questionnaire survey and included information on their age, gender, educational level, farming level, household size, farming years, average monthly income and percentage of income from farming. The following four major respondent age groups were considered: 20–29, 30–39, 40–49, 50 and above. Most respondents (87%) in Riyom were within the older age groups of 40 and above; few (13%) within the younger age groups were engaged in farming activities. However, in Shendam, there was a near even distribution (30%) in each of the age groups with the exception of the youngest (20–29) age group, which accounted for almost 10%. The relative proximity of Riyom to Jos, the state capital, perhaps explains the low

number of younger individuals in farming activities. The ratio of male to female respondents was approximately 7:3, with a higher proportion of female farmers in Riyom. The average household size was 7 persons in Riyom and 8 persons in Shendam. In terms of their education level, 65% of respondents on average attained the basic literacy level; Riyom, however, had a lower value with 43% below the literacy level. On average, 57% of respondents are full time farmers, and more than half of the farmers had other non-farm sources of income. Although this study did not consider the individual crop area under cultivation, it found that most farmers (68%) in Shendam made over 50% more in income from farming than their counterparts in Riyom. These results informed knowledge on the backgrounds of respondents.

## 5.2. Indicators of Climate Change

This section presents an analysis of the local indicators of climate change based on farmers' perceptions of climate risk in the area. Farmers' knowledge of the changes in average weather conditions over their years of farming experience were considered. In both communities, most respondents (91%) had over 10 years' experience in farming, which is considered sufficient to recount opinions for the observed weather changes over time. Although a minimum of 10 years farming experience was the benchmark sufficient for farmers to observe a meaningful change in the climate, respondents with less than 10 years farming experience (9%) were not excluded in the analysis, since their knowledge of the area as residents for over 20 years was found to be relevant. All respondents (100%) supported evidence of climate change in several ways; both related to warmer and drier patterns as well as shifts in rainfall patterns. These results are presented in Table 3.

**Table 3.** Local Indicators of Climate Change.

| Local Changes | Indicators | Percent (%) | Percentage Scores * | |
| --- | --- | --- | --- | --- |
| | | | **Shendam** | **Riyom** |
| Warm and Dry Patterns | Reduced stream flow | 100 | 10.0 * | 10.0 * |
| | Rises in temperature | 97 | 9.1 * | 10.0 * |
| | Drying of wetlands | 89 | 8.3 | 9.3 * |
| | Longer dry periods | 87 | 8.3 | 9.1 * |
| | Prolonged dry spells | 83 | 7.8 | 8.7 |
| | Water shortages | 53 | 5.2 | 6.8 |
| Rainy and Wet Patterns | Heavier rains | 100 | 10.0 * | 10.0 * |
| | Destructive winds | 98 | 9.6 * | 10.0 * |
| | Irregular rains | 95 | 9.6 * | 9.5 * |
| | Less rain days | 92 | 9.1 * | 9.3 * |
| | Late onset of rains | 92 | 8.7 | 9.6 * |
| | Early cessation of rains | 89 | 8.7 | 9.1 * |
| | More floods | 78 | 10.0 * | 6.4 |
| | Destructive hail | 63 | 3.9 | 9.1 * |

* High risk indicators = 90th percentile.

Indicators of changes towards warm and drier patterns included reduced stream flow (100%), rises in temperature (97%), the drying of wetlands (89%), longer dry periods (87%), prolonged dry spells during rainy seasons (83%), and water shortages (53%). Furthermore, indicators of changes in rainfall patterns included experience of heavier rains (100%) leading to flooding (78%), irregular rains (95%), the late onset of rain (92%), destructive wind (98%), hail storms (63%), the early cessation of rains (89%), and fewer rainy days (92%). The results in Table 3 reveal that 100% of respondents agreed that reduced stream flows and heavier rains are the main local indicators of climate change. Moreover, farmers suggest that the least significant indicators are water shortages and destructive hailstorms. Although these findings provide state averages, the study identified local differences between locations and found variations between individual communities.

After assigning percentage scores to indicators and selecting high risk indicators (90th percentile range) in each community, Shendam records 7 high risk indicators: heavier rains (10.0), floods (10.0), reduced stream flows (10.0), destructive winds (9.6), irregular rains (9.6%), rises in temperature (9.1), and fewer rainy days (9.1). Riyom records a higher number for 11 high risks: heavier rains (10.0), destructive winds (10.0), rises in temperatures (10.0) and reduced stream flow (10.0). Others include the late onset of rain (9.6), irregular rain (9.5), the drying of wetlands (9.3), fewer rainy days (9.3), drier periods (9.1), the early cessation of rain (9.1), and destructive hail storms (9.1). This confirms the wide distribution of seasonal variation on the Plateau highland where Riyom is located. When respondents were asked to quantify the effects of each climate risk on an impact scale, all respondents perceived that at least one climate risk was disruptive. Figure 3 presents a summary of the farmers' perceptions of the impact levels of climate risk in the two case study communities. The distribution of respondents' opinions is indicative of the prevalent climate-related event experienced in each location.

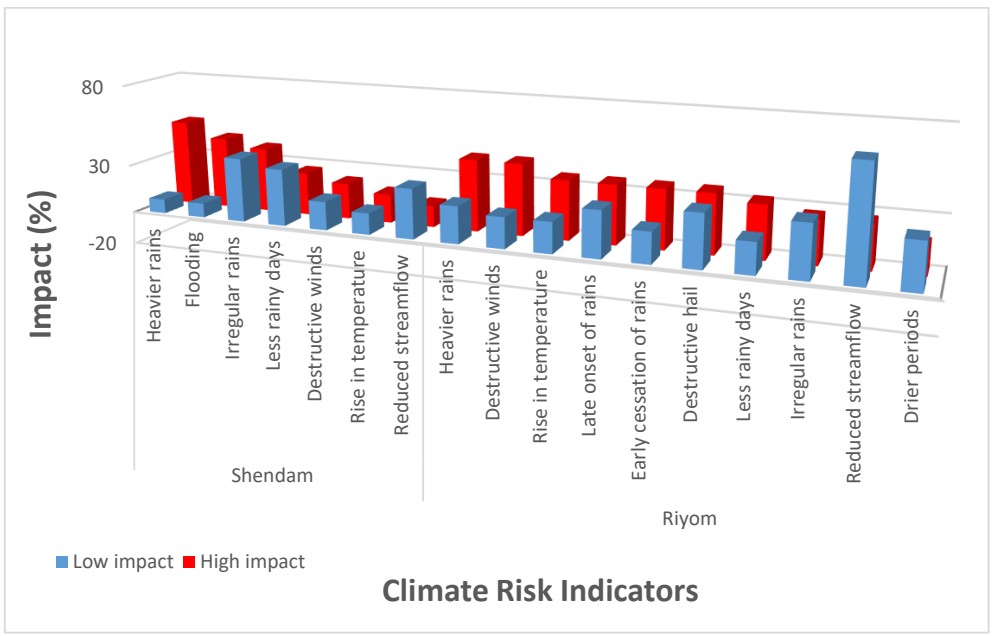

**Figure 3.** Farmers Perception of Climate Change Risk in Shendam and Riyom. Impact level of climate risk is expressed as 25th percentile = low impact and 75th percentile = high impact.

High impact (75th percentile) and low impact (25th percentile) risk levels revealed local differences in the climate risks between Shendam and Riyom. High impact climate risks in Shendam concentrated on wet patterns, which translate to a higher chance of flooding, according to the key informants. Furthermore, high risks in Riyom encompass a combination of alterations in both dry and wet conditions. Although Riyom experiences a wide variation of weather conditions, the area's geographical terrain, as explained in Section 4.1, is located on a high elevation, and is less likely to record frequent floods. Evidence of dry conditions (rising temperatures, the late onset of rains, the early cessation of rains and fewer rainy days) are found to contribute to the spread of plant diseases, leading to water shortages and, in turn, agricultural droughts, as indicated by the interview findings.

*5.3. Impacts of Climate-Related Events on Agrarain Infrastructures*

Understanding the local impacts of climate-related events could help communities prepare for future uncertainties. Even though the study revealed variations in the local indicators of climate change and, in turn, the climate-related events experienced in the study locations, all respondents acknowledged exposure to at least one type of climate risk event. The main climate-related events identified from the literature and prevalent in the area are floods, extreme temperatures, droughts and storms. The impact assessment was performed using a rank based approach, which enabled

respondents to assign relative quantities to opinions based on their experience and local knowledge. Respondents were asked to rank their opinions on a scale of 1 to 5 where 1 = 'no impact' and 5 = 'high impact'. They were asked to rate the effect of the identified climate events on the various components of agrarian roads in Shendam and the irrigation infrastructures in Riyom. The findings indicate that climate-related events affect agrarian infrastructure systems on a variable scale, and this is due to differences in location, the scale of the climate risk and the level of infrastructure exposure to the identified risk. Table 4 presents the results of the impact levels of climate-related events on infrastructure systems based on mean scores of farmer's responses.

**Table 4.** Perceived Impacts of Climate-related events on Infrastructure Components.

| Elements Affected | Climate Events/Impact Level [1] | | | |
| --- | --- | --- | --- | --- |
| | **Floods** | **Extreme Temperatures** | **Droughts** | **Storms** |
| **Case 1: Agrarian Road in Shendam** | | | | |
| Road pavements | H | VL | VL | VL |
| Bridges | H | VL | VL | VL |
| Culverts | H | VL | VL | VL |
| Drainage | H | VL | VL | VL |
| **Case 2: Irrigation Systems in Riyom** | | | | |
| Small earth dams/water catchment | VL | L | M | VL |
| Boreholes | VL | L | M | VL |
| Tube wells | VL | L | M | VL |

Results of the impact analysis on agrarian roads in Shendam indicate that floods are high risk events affecting all road infrastructure components, while extreme temperatures, droughts and storms have the least impact on rural roads. The findings on the impact of climate-related events on irrigation systems in Riyom indicate that droughts record the greatest impacts. Findings also indicate that due to the low probability of occurrence, floods had very low impacts on irrigation infrastructures in Riyom. Even though rain and wind storms are high probability events in Riyom, as indicated by the findings on the local indicators of climate change, the analysis of the results shows that it has a very low impact on irrigation infrastructure systems. Extreme temperatures were identified as having a relatively low impact on irrigation systems. In comparing the impacts of various climate-related events on infrastructure types, an in-depth understanding of how future climate change will affect agrarian infrastructures can be incorporated into any adaptations and infrastructure resilience planning. The next sections discuss the findings of each case study location.

5.3.1. The Impacts of Floods on Agrarian Roads in Shendam

Shendam, located on the lowland, is susceptible to frequent floods, and road infrastructures are susceptible to damage due to the poor nature of agrarian roads in the area. The intensity of rainfall is identified as a factor that increases the probability of flooding and surface runoff. Findings from the questionnaire survey indicate that floods account for the highest impact of climate-related events on agrarian roads, damaging road pavements, affecting bridges, washing out culverts and eroding drainage. All respondents (100%) agreed that floods in Shendam had devastating impacts on agrarian roads. The mean scores show that respondents indicated that all components of the roads under study in Shendam, including road pavements (4.9), bridges (4.8), culverts (4.8) and drainage (4.8), suffered high damage due to floods. Extreme temperatures, droughts and storms had insignificant effects due to the low probability of occurrence; this was corroborated by the interview findings.

In-depth information from key informants was used to corroborate the farmers' responses and determine the extent of any infrastructure damage. Thus, findings from the qualitative analysis

---

[1]    Impact level: H = High impact, M = Moderate impact, L = Low impact, VL = Very Low impact.

indicated that heavy rains, which lead to frequent floods, are experienced in the area almost annually. Participants recounted that the floods in 2012 alone comprised a major flood event. This was likened to a once-every-40-years occurrence when analyzing the impacts of climate-related events driven by climate change. One participant explained that:

> *"Our highest problem in Shendam is flood. In the year 2012, several days of heavy downpour caused floods and almost all villages within Shendam suffered. The Shendam town bridge was totally washed away, and two other bridges in the surrounding villages collapsed cutting off the villages". (I$_{09}$)*

Participants further indicated that floods have become more frequent and devastating in the last 10 years, with at least one occurrence recorded each year. In recounting the extent of the flood damage in Shendam, one participant explained:

> *" . . . serious challenges of flood in Shendam. I think in the entire Plateau state, Shendam is the worst hit by floods. We experience serious floods, which affect our farmers, their farmlands, houses, and some infrastructures. More than 100 hectares of farmland were flooded. Roads connecting to the riverine areas are affected and even cut off. Even within Shendam town, the bridge linking Shendam and Jos road was cut off during the 2012 flood. The bridge linking Shendam to Yelwa was cut off, and the bridge linking Shendam to Kalong was cut off. Three bridges in Shendam were cut off that year. So the people resorted to using canoes". (I$_{06}$)*

Key informants generally acknowledged that heavy rains and overflowing rivers are the primary causes of floods in Shendam area. One interviewee explained that three days of heavy rainfall led to devastating floods which affected road transportation system in the area. Three main bridges were heavily damaged. Major roads linking the local government headquarters and other villages were badly affected. Various levels of damage to road surfaces, road embankments and culverts were recorded. Communities had access challenges as transport services suffered disruptions. Commuters were left with no option other than to take longer routes at higher prices or to crossover to Shendam town by canoes. Interviewees, who are infrastructure managers with technical backgrounds, identified the vulnerabilities arising from the proximity to hazard sources and the high risk of a hazard occurrence. They explained that, due to Shendam's low elevation and the recurrent incidence of floods in the area, a higher priority is given for infrastructure rehabilitation in times of need. Furthermore, participants explained that, although floods of different magnitudes were experienced over the years, they were yet to fully recover from the 2012 flood, as parts of the damaged roads were still in disrepair. Although most damaged parts had undergone reconstruction or rehabilitation, the majority of farmers (79%) said that the damage was a major setback to farming activities, while others (21%) said it had no effect on them.

### 5.3.2. Impacts of Droughts on Irrigation Systems in Riyom

Agricultural droughts are the types of drought identified in Riyom. These result from insufficient moisture to meet crop needs at a particular time, alongside hydrological droughts due to shortages in the supply from surface and sub-surface water. Indeed, 70% of farmers in Riyom engage in both dry and rainy season cultivation for year-round production, relying on natural streams and small earth dams to irrigate their crops. Farmers occasionally resort to wells (48%) and borehole sources (41%) to augment supplies during periods of shortage within the peak of the dry season. While 58% of respondents agree that water shortages represent challenges for irrigation farming, 34% attribute this to a rise in temperature. Although 92% of respondents agreed that there are drier conditions in the area, only 68% consider this as drought. The results of the mean scores indicate the impact of droughts on irrigation systems and indicate a moderate impact on earth dams and natural streams (3.2), boreholes (3.4) and wells (3.6). Findings also indicate that warmer temperatures influence drought conditions as the results show for earth dams/streams (2.3), boreholes (2.4) and wells (2.4). Furthermore, results suggest that floods and storms have insignificant impacts on irrigations systems.

Using the findings from key informants to corroborate the farmers' responses, the results reveal that droughts driven by water shortages are apparent in the area. One interviewee explained that

*" . . . in areas where irrigation takes place, the source of water around that area normally lasts up to January- February, but I don't know what happened . . . before we knew it, by early December the water dried up. It was a very serious problem and of course there was no magic we could do". (I_{19})*

Findings from the in-depth interviews reveal changes in the seasonal stream flows. Perennial streams experience less water volume, while seasonal streams are dry. A participant recounted how streams are now dry by about 2 months before the expected date.

*" . . . in recent times, farmers who engage in dry season farming will definitely encounter drought which then deters the growth of the crop" . . . "We tried all we could, in fact we sank two boreholes just to augment but eventually we lost a large chunk of the farm because there was really no water . . . ". (I_{18})*

Whilst another interviewee explained that *" . . . sometimes farmers have to sow seeds more than once because of water shortages" (I_{05})*.

Participants recognized that the erratic rainfall experienced in the area in recent years could be the reason for stream flow changes. However, some stated that it was difficult to differentiate between water shortages due to intensive use and water shortages due to changing weather patterns. However, most participants agreed on the slight rise in temperature, which contributed to a higher evaporation rate and placed higher demands on farmers when irrigating their crops. Findings also indicate that lower water levels affect the yields of water sources, which, in turn, challenge the potential for irrigation farming in the area. Moreover, quick interventions, such as accessing alternative supplementary sources of water during periods of shortfall reduced farm losses. Furthermore, farmers who devised intermediate strategies, such as spending more money to source for water or replanting seeds after periods of shock, were able to sustain production, although these are dependent on the farmer's financial capacity.

*5.4. The Cascading Effect of Infrastructure Disruption on Agrarian Livelihood Systems*

Having identified the direct impacts of climate-related events (floods and drought) on agrarian infrastructure, this section discusses the indirect impacts, also referred to cascading effects. This focuses on how agrarian infrastructure disruption in the two case study communities constrained agricultural production, the rural economy and human activities. These culminate in agrarian livelihood systems.

5.4.1. Cascading effect of Floods

Although physical damage to road network systems and the disruption of services are identified as the direct impacts of floods in Shendam and listed below, a chain of negative effects are identified and summarized in Appendix B.

- Agricultural activities: In explaining how the damage to roads affected farming activities, farmers highlighted that, apart from the physical destruction of farmlands and crops, the loss of transport services made it almost impossible to transport inputs, such as fertilizers to farming communities, and crops from farm to market. This led to large amount of crop waste, particularly amongst perishable crops. Transport fares doubled and road damage alone accounted for about 50% of the crop waste. These are, however, estimates based on farmers' responses, and not actual figures.
- Rural economic activities: Farmers also noted that the time of the disaster event coincided with the peak of the rainy season when farmers often moved food crops from barns to market in order to take advantage of the peak price periods, as more profits are made at such times. These difficulties contributed to the low returns on farmers' investments, and in turn their income levels. Farmers explained that due to the loss of crops and low-income levels, the following farming sessions

were affected, as they lacked the capacity for intense cultivation following huge losses from the previous year. Participants noted a general rise in the prices of goods, for both food crops and non-food items, around the study area after the event. This was attributed to the flood; however, it was difficult to separate the goods from areas genuinely affected from those taking advantage of the situation. Also, commercial activities and local revenue generation on market days were affected. The usual local tax collection and toll gate fares from traders and motorists on market days were low, thereby affecting the local economy.

- Human activities: respondents explained that losses from both crop damage due to the flood waters and crop waste due to transportation disruption caused psychological stress for large scale farmers. The livelihood sources of farmers without insurance were lost, which accounted for an increase in the poverty levels and a heightened food crisis. One respondent maintained that, due to the bridge collapse, there was also a temporary loss of leisure activities, as it was difficult to access the town center.

While most farmers (79%) agreed that infrastructure affected farming and other agrarian activities, others (21%) said it had no effect. In order to understand why the respondents' opinions on flood impacts varied, the study analyzed the factors influencing farmers' perceptions of the flood impacts. These results are presented in Table 5.

**Table 5.** Factors Influencing Farmer's Perceptions of Flood Impacts[2].

| Explanatory Variables | B | S.E. | Wald | df | Sig. | Exp (B) | 95% C.I. for EXP (B) | |
|---|---|---|---|---|---|---|---|---|
| | | | | | | | Lower | Upper |
| Location | −22.77 | 4249.98 | 0.00 | 1 | 0.996 | 0.00 | 0.00 | |
| Age | 2.58 | 0.79 | 10.79 | 1 | 0.001 | 13.24 | 2.84 | 61.82 |
| Gender | 0.37 | 0.53 | 0.48 | 1 | 0.490 | 1.44 | 0.51 | 4.06 |
| Education level | 1.04 | 0.61 | 2.95 | 1 | 0.086 | 2.83 | 0.86 | 9.26 |
| Farming years | 0.14 | 0.97 | 0.02 | 1 | 0.888 | 1.15 | 0.17 | 7.72 |
| Income level | −0.41 | 0.84 | 0.24 | 1 | 0.623 | 0.66 | 0.13 | 3.42 |
| Percentage of farm income | −2.05 | 0.57 | 13.12 | 1 | 0.000 | 0.12 | 0.04 | 0.39 |
| Constant | 20.96 | 4249.98 | 0.00 | 1 | 0.996 | 1,266,182,881.51 | | |

Using a binary logistic regression to analyze the factors affecting the likelihood of farmers agreement on the flood impacts, results shows the model was statistically significant [$X^2$ (7, $n$ = 175) = 78.343, $p < 0.05$]. The full model contained 7 independent variables (location, age, gender, educational level, farming years, income level, and percentage of farm income). The whole model correctly classified 79.4% of cases, which explained between 36.1% (Cox and Snell $R^2$) and 56.6% (Nagelkerke $R^2$) of the difference in farmers' perceptions of the flood impacts on agrarian roads. Table 5 shows the logistic regression coefficient (B), Wald test, and odds ratio (Exp.B), among others, for each of the explanatory variables. While the Wald Chi-square statistics tests the unique contribution of each explanatory variable, the odds ratio represents the contact effect of a variable on the likelihood of an outcome. Utilizing a 0.05 criterion for statistical significance, age of respondents ($p = 0.001$) and percentage of farm income ($p = 0.000$) are statistically significant. From the results of the odd ratio in Table 5, the age (13.24) which is the highest predictor of the model, shows that when holding all other variables constant, respondents from the older age ranges (above 40 years) are 13 times more likely to report the flood impacts. So also significant, the effect of percentage of farm income is much smaller than that of age of respondents, with the odds of reporting a high impact of flood by a multiplicative factor of 0.12. Although non-significant, the odds ratio of educational level (2.83), gender (1.44) and farming years (1.15) are notable. In addition to non-significant variables but in explaining the direction

---

2    $Y = bo + biXi + biiXii + \cdots bnXn$)where $Y$ is impact status (1 = impact, 0 = no impact).

of the relationship (B), the location (−22.77), percentage of income (−2.05) and income level (−0.41) revealed negative relationships, while educational level (1.04), gender (0.37) and farming years (0.14) indicated positive relationships.

Although famers agreed that government interventions, such as seedlings, fertilizers and agrochemicals, were provided as supplements for losses, these were insufficient for the recovery effort required considering the extent of the farmers' losses. Key informants explained that although the initial cause of infrastructure damage was the intense flood, the condition of the infrastructure at the time of the event contributed to the extent of the damage. Agrarian roads are generally in a poor condition; they are poorly constructed and lack regular maintenance. Furthermore, in explaining the current conditions of agrarian roads, interviewees identified four main factors contributing to the poor infrastructures conditions. These include financial constraints (83%), management constraints (67%), political constraints (50%) and environmental constraints (25%). Financial, managerial and political constraints are institutional factors, while environmental factors concerned challenges with the terrain of the area. Participants considered whether less damage would occur if both structural and institutional measures were in place to provide and manage infrastructure assets. Whereas little can be done to prevent the occurrence of extreme events like floods, the provision of a strong infrastructure can lessen the extent of the damage by such events. Participants noted that, due to financial constraints on the reconstruction of damaged infrastructure assets shortly after the floods, the protracted loss of transport services to affected areas caused further human and economic losses. A summary of the impacts of floods on AI and its cascading effect in Shendam is presented in Appendix B.

### 5.4.2. The Cascading Effect of Drought

Although drought conditions caused by changing temperatures and rainfall patterns have direct impacts on irrigation systems, the multiple effects of infrastructure failure can follow in sequence. These are effects on crop production, rural economic activities and human activates which culminate in agrarian livelihood systems.

- Agricultural activities: The impact of irrigation disruption on crop production can be summarized as 'low water yields, poor crop yields'. Furthermore, planted seed and applied agrochemicals are wasted, whilst plant pests/diseases spread, and eventually there is a loss of operation. These add to the financial implications for farmers and the community as a whole.
- Rural economic activities: Farmers sometimes incur additional costs to sustain irrigation farming as they tend to spend more on labor to irrigate their crops. Farmers often spend more money to dig wells several meters deep to source for water and to fuel motorized pumps in order to irrigate crops. At other times, when the water crisis is severe and beyond farmers' capacities, the authorities provide immediate alternatives, such as the construction of boreholes to minimize damage due to the harsh conditions. However, this is not the case at all times. At the end of the farming season, farmers sometimes record low returns on investment after spending huge sums of money to procure labor, and face the challenge of 'middle men', who largely determine the market prices of food crops. Although unstructured market prices are a deterrent to farmers, they are obliged to sustain production, which is still considered a 50 percent win.
- Human activities: Due to overcrowding and competition amongst various water users, there are cases of conflict, particularly between farmers and herdsmen, over the control of space and water. The destruction of crops and livestock, a loss of trust, the loss of livelihoods and eventual migration are noted as the results. Poor crop yields occur due to water scarcity alongside the destruction of crops and livestock due to conflicts, which tend to worsen the food crisis. In a bid to maintain law and order in crisis communities, a respondent explained that the local government is now compelled to redirect the limited funds meant for infrastructural development to maintain additional security services within the local government area. In their opinion, peace and security are top priorities over infrastructural development, as farmers need a clear environment to grow

crops. Therefore, this is in agreement with the assertion of Al Khaili et al. (2013) that disasters can have a direct or indirect impact on the environment.

However, interviewees suggested that quick interventions, such as accessing other supplementary sources of water during periods of shortfall, reduce farm losses. Alternatively, farmers can devise intermediate strategies, such as replanting seeds after periods of shock, or spending more to source water in order to sustain production. However, this is dependent on a farmer's capacity. Despite informants' opinions on the impact of droughts, and farmers' (72%) confirmation of the cascading effect of infrastructure disruption, 28% respondents revealed that infrastructure disruption had no effect on agrarian activities. In order to understand why farmers' perceptions towards drought varied in the area, the study used a binomial logistic regression model to analyze factors that were likely to influence the farmers' perceptions of drought impacts in Riyom. The results are presented in Table 6.

**Table 6.** Factors Influencing Farmer's Perception of Drought in Riyom.

| Explanatory Variables | B | S.E. | Wald | df | Sig. | Exp (B) | 95% C.I. for EXP (B) | |
|---|---|---|---|---|---|---|---|---|
| | | | | | | | Lower | Upper |
| Farming Season | 0.39 | 0.65 | 0.36 | 1 | 0.547 | 1.48 | 0.41 | 5.33 |
| Age | 2.31 | 0.77 | 8.99 | 1 | 0.003 | 10.03 | 2.22 | 45.25 |
| Gender | −0.95 | 0.54 | 3.10 | 1 | 0.078 | 0.39 | 0.14 | 1.11 |
| Education level | 0.58 | 0.66 | 0.78 | 1 | 0.377 | 1.79 | 0.49 | 6.51 |
| Farming years | −0.08 | 1.07 | 0.01 | 1 | 0.943 | 0.93 | 0.11 | 7.58 |
| Income level | −0.76 | 0.814 | 0.87 | 1 | 0.352 | 0.47 | 0.10 | 2.31 |
| Percentage of farm income | −1.16 | 0.60 | 3.77 | 1 | 0.053 | 0.31 | 0.10 | 1.01 |
| Constant | 0.41 | 1.61 | 0.06 | 1 | 0.800 | 1.51 | | |

The results of the binomial logistic regression analysis of the factors influencing farmers' perceptions of drought impacts confirm the statistically-significant model [$X^2$ (7, $n = 106$) = 26.43, $p < 0.05$]. While explaining between 23.25% (Cox and Snell $R^2$) and 32.9% (Nagelkerke $R^2$) of the differences in respondents' perceptions, the model correctly classified 78.0% of the cases. Using a 0.05 criterion for statistical significance, age of respondents ($p = 0.003$) was significant. From the results of the odd ratio in Table 6, the age (10.03) which is the highest predictor of the model, shows that when holding all other variables constant, respondents from the older age ranges (above 40 years) are 10 times more likely to report the impacts of drought. Although non-significant, results of the odd ratio shows that educational level (1.79) and farming season (1.48) recorded notable figures which could have influences on explaining farmers' perceptions of droughts. Percentage of income records a borderline of $p = 0.053$. Also non-significant, the odds ratio of farming years (0.93), income level (0.47) and gender (0.39) was much smaller and their coefficient (B) depicts a negative direction of relationship.

These models limit the explanatory variables of farmers' socio-economic backgrounds based on an understanding of the relationship between their socioeconomic status and capacity outcome in the literature. Further research will explore farmers' coping strategies, the community's adaptive capacities and the institutional response capacities to climate change and climate-related hazards.

*5.5. Implications for Farming Communities*

The damage to agrarian roads and the failure of irrigation systems contribute to the disruption of agrarian activities. Farmers, the main infrastructure users, can record huge losses, leading to a decline in rural economic activities. Low yields and huge farm losses affect the supply chain, as farmers are unable to provide the required quantity of crops to major marketers. This also impacts local transporters who convey crops from communities to markets. Furthermore, low supplies mean a general rise in the price of goods on the market. These present several challenges including increased spending for the rural populace, a rise in poverty levels, a threat to the national food security, and a decline in the sector's GDP contribution.

Famers with a low socio-economic status generally lacked the capacity for rapid recovery from losses. Due to losses in the agrarian livelihood systems, farmers had to adjust their life patterns, e.g., by consuming fewer meals, spending less and engaging less frequently in social/extra-curricular activities. This meant a higher likelihood of hunger and malnutrition, particularly amongst older people, pregnant women and children. However, a positive side is the diversification of income and the diversification of cropping patterns. Farmers with alternative sources of income adapted better. Family assets, such as vehicles, livestock and personal belongings, were sold in order to sustain the family, recover from losses and resume crop production. This, however, left them feeling impoverished and insecure. Other effects include psychological discomfort and the loss of life; these were common with major farmers who invested huge amounts in farming activities.

*5.6. Policy Implications*

Climate change and its impacts have continuously been a challenge for the Nigerian agricultural sector. The local indicators of climate change are evidence that the increasing occurrence of climate-related events will have implications on the durability of agrarian infrastructure for sustainable agricultural production. Both agrarian roads and irrigation infrastructure are insufficient, and what little that is available is continuously under threat from changing weather patterns. Adverse flood events, damaged road systems and transport service disruption, are usually exacerbated by the poor conditions of road facilities. The current decade showed the highest level of road and bridge damage by climate-related events in the recorded history of Plateau state. Furthermore, droughts are having a negative impact on the surface water for irrigation and other agrarian activities, leading to a chain of events that affect communities and the economy in general. The Riyom local government area unfortunately falls amongst the lowest ranking in terms of its infrastructural development; as such, the area lacks large irrigation facilities, such as dams, to harvest excess water during the rainy season and harness its dry season potential. Low yields from boreholes, particularly at the peak of the dry season, alongside the ephemeral nature, suggest a lack of proper construction.

The projection of more adverse climate events will mean additional pressure for infrastructure development and reconstruction. It is therefore necessary to address the challenges of the AI gap by providing new structures and retrofitting current transportation facilities that can withstand adverse weather conditions. If not addressed, infrastructure damage and service disruption will increase as the frequency and intensity of climate-related events are expected to double. A major challenge when addressing climate change impacts on developing countries, such as Nigeria, includes the lack of planning for uncertainties. Policies and plans are tailored towards solving immediate challenges, popularly called damage control. For instance, road rehabilitation is not often approved unless there is critical damage to the facility leading to accidents and the loss of life; when this occurs, road rehabilitation can take place. This is due to a lack of funds, low budgetary allocations, and/or the misappropriation of funds. Government regulations can help to safeguard against the provision of low quality infrastructure. This can, for instance, be achieved by holding officers/infrastructure providers to account for projects that are executed below standard during their tenures. Also, they can ensure that the total value of a piece of infrastructure at the point of delivery is commensurate with the amount awarded for its construction.

## 6. Conclusions and Recommendations

This study revealed that the adverse effects of climate-related events go beyond disturbances to the agrarian infrastructure to affect other non-farm agrarian livelihoods. Although previous studies have evaluated the impacts of climate change on agriculture in Nigeria (Adegoke et al. 2014; Akpodiogaga-a and Odjugo 2010; Morton 2007; Schlenker and Lobell 2010), this study presents a new dimension to evaluate agricultural losses due to the disruption of agrarian infrastructure. In this regard, future research can adopt this means of identifying a cascading effect to quantify the indirect impacts of climate change.

Besides integrating local knowledge with institutional perspectives, the approach provides a suitable process to supplement data sources. Farmers are perceptive of climate change as the driver of climate events, its impacts on agrarian infrastructure and the effects on agrarian livelihood systems. Findings reveal the local indicators of climate change, namely, alterations in rainy patterns and warm-dry conditions, and increases in the occurrence of floods and drought. Most respondents agreed that at least one of the following had a chain effect on agrarian livelihoods: damage to roads and network systems, transport service interruption, low water yields from irrigation infrastructure or irrigation interruption. The direct impacts of floods on agrarian roads include damage to road surfaces, damage to bridge pillars and roads leading to bridge collapses, and a total washout of bridges and culverts. The indirect impacts on agricultural production include a loss of production, food waste, increases in the cost of transportation and inputs. The cascading effect on economic activities includes market instability, the low patronage of small scale industries, the low return on investments, and disruption to commercial activities due to supply chain disturbance and general constraints on economic development. Other cascading effect and human impacts include the loss of life and livelihoods, displacement, the spread of disease and epidemics, an increase in poverty levels and food security, and the disruption of social activities. Secondly, the direct impact of droughts on irrigation facilities includes low water levels leading to the low yield and quality of water sources. The effect of this on agricultural production includes low crop yields, wasted inputs, such as seeds and agrochemicals, and the loss of production. The cascading effect and economic impacts include low returns on investment due to the high cost associated with accessing irrigation facilities and sourcing water for irrigation, and the disruption of commercial activities due to the inability to sustain a supply. Other cascading effects on human activities include overcrowding and competition for water sources leading to strife and conflict, the loss of human life, trust and livelihoods, human displacement, an increase in poverty levels, food insecurity, and the pressure on authorities and security agencies. Further findings from the study suggest that the condition of infrastructure at the time of the climate-related event exacerbate the impacts, as poor conditions make infrastructure vulnerable. The four major factors found to influence the status of infrastructure are (i) institutional management, (ii) financial factors, (iii) political factors, and (iv) structural factors. Therefore, further research on the factors affecting infrastructure conditions is recommended in order to employ both structural and institutional measures as a means for risk reduction and to improve the resilience of agricultural infrastructure.

**Author Contributions:** Both authors contributed equally to this work.

**Funding:** This research received no external funding.

**Conflicts of Interest:** The authors declare no conflict of interest.

## Appendix A

**Table A1.** Profile of Respondents.

| (A) Profile of Key Informants | | Count (*n* = 12) | Percent (%) |
|---|---|---|---|
| | Federal | 2 | 16.7 |
| Administrative Level | State | 4 | 33.3 |
| | Local/Community | 8 | 76.7 |
| | Technical | 4 | 33.3 |
| Background | Planning | 6 | 50.0 |
| | Mobilize/supervision | 4 | 33.3 |
| Gender (%) | Male | 10 | 83.3 |
| | Female | 2 | 16.7 |
| Years of experience (mean) | | 18.6 | |

**Table A1.** *Cont.*

| (B) Profile of Farmers Surveyed in Shendam & Riyom | | Total (*n* = 175) | Shendam (*n* = 69) | Riyom (*n* = 106) |
|---|---|---|---|---|
| Age groups (%) | 20–29 | 5.7 | 8.7 | 3.8 |
| | 30–39 | 17.7 | 30.4 | 9.4 |
| | 40–49 | 37.1 | 30.4 | 41.5 |
| | 50> | 39.4 | 30.4 | 45.3 |
| Gender (%) | Male | 70.3 | 91.3 | 56.6 |
| | Female | 29.7 | 8.7 | 43.4 |
| Education level (%) | Primary | 19.4 | 8.7 | 26.4 |
| | Secondary | 28.6 | 43.5 | 18.9 |
| | Tertiary | 36.6 | 34.8 | 37.7 |
| | Informal | 15.4 | 13.0 | 17.0 |
| Farming level (%) | Full time farmer | 56.6 | 39.1 | 67.9 |
| | Part-time farmer | 43.4 | 60.9 | 32.1 |
| Household size (mean) | | 7.56 | 7.99 | 7.28 |
| Farming Years (%) | <5 | 2.9 | 4.3 | 1.9 |
| | 5–10 | 6.3 | 4.3 | 7.5 |
| | >10 | 90.9 | 91.3 | 90.6 |
| Average monthly income (%) | <15,000 | 26.3 | 8.7 | 37.7 |
| | 15,000–50,000 | 46.3 | 39.1 | 50.9 |
| | >50,000 | 27.4 | 52.2 | 11.3 |
| Percentage of income from farming (%) | 25 | 29.1 | 4.3 | 45.3 |
| | 50 | 22.3 | 27.5 | 18.9 |
| | 75 | 41.1 | 55.1 | 32.1 |
| | 100 | 7.4 | 13.0 | 3.8 |

# Appendix B

**Table A2.** Cascading Effects of Agrarian Infrastructure Disruption.

| List of Cascading effect of Agrarian Infrastructure Disruption | |
|---|---|
| Shendam | Riyom |
| Impacts of Climate-related events on Agrarian Infrastructure | |
| Impacts of floods on road network system | Impacts of drought on irrigation systems |
| -Washout of bridges and culverts<br>-Washout of bridge and road embankments<br>-Damage to road surfaces<br>-Disruption of transport services | -Low water levels<br>-Low yields of dams, boreholes and wells<br>-Low water quality |
| Cascading effect on Agrarian Livelihoods | |
| Agriculture | |
| -Inability to access farms, communities and markets<br>-High cost of transportation<br>- High cost of inputs: fertilizer, seeds<br>-Waste of food crops<br>- Inability to meet demand<br>-High loss and low profit<br>-Loss of production due to infrastructure damage | -Poor crop yields<br>-Waste of inputs: seeds and agrochemicals<br>-Spread of plant pests and diseases<br>-Loss of crops<br>-Loss of production due to low water levels affecting irrigation infrastructure |
| Rural Economic Activities | |
| -Market instability and Price hike of goods<br>-Low patronage of small scale industries: rice mills<br>-Disruption of commercial activities due to supply chain disruption | -High cost of sourcing water<br>-Cost of constructing alternative irrigation facilities<br>-Less profit<br>-Disruption of commercial activities due to inoperability |
| Human Activities | |
| -Loss of human lives<br>-Loss of livelihoods<br>-Human displacement/ temporary migration<br>-Emotional and psychological effects<br>-Increase in poverty levels<br>-Food crisis<br>-Disruption of social activities | -Overcrowding and competition on water sources<br>-Strife and conflicts<br>-Loss of trust<br>-Loss of human lives<br>-Loss of livelihoods<br>-Human displacement<br>-Increase in poverty levels<br>-Food crisis<br>-Pressure on authorities and security agencies |

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
