# Peer review of "Farmers Perceptions of Climate Change Related Events in Shendam and Riyom, Nigeria"

_economies, doi:10.3390/economies6040070_

Round 1
Reviewer 1 Report
The language of the paper is clear and lucid and messages are appropriately captured.
General comments:
1. The title of the paper is broader than its content. While the title focuses on the impact of climate change, the content is on the impact of floods and droughts. Floods and droughts are not climate change but its manifestations. So, the title should be revisited to reflect the impact of floods and droughts.
2. The value added of the paper has not been articulated. It is therefore difficult to know its contribution to knowledge.
3. The analytics of the paper is static. The focus of the paper requires some dynamic and trend analysis showing the short and long-term trends of the issues - indicators of climate change, floods and droughts.
4. The policy implication of the paper is yet to be fully addressed.
Specific comments
5. The abstract should be refocused to address the rationale and value add of the paper, objectives, findings and policy implications.
6. Section 1 brings out the rationale and the value added of the paper.
7. Section 2 focuses on definitions or conceptual issues. An overview of the trends, scope and magnitude of agricultural infrastructure in Nigeria and possibly in Plateau state is needed to contextualize the paper and to help establish the import of the paper. There are conflicting statistics in the paper. For instance, lines 8 and 99 provide different statistics (70% and 80%) for the proportion of the population employed in the agriculture. The reference year is also absent.
8. Section 3 The analysis here should be beefed up. The analysis of the various events (floods and droughts) and trends over time deserve attention. It is too static. Why are floods and droughts an issue in Plateau State, Nigeria? Issues on lines 173-180 and table 3 should be addressed in this section.
9. Section 4 as it is there is no methodology. How did you select the interviewers and the key informants? How many respondents were used for the structured questionnaires? Using 8 key informants for the open-ended questions seem inadequate for this topic. What research method did you use for the analysis – qualitative or quantitative? Given the title of the paper, there should a regression analysis showing the impact of floods and droughts.
10. Section 5 the issues discussed here are generally known facts. The paper should tell something new. The issues provided in tables 4 and 5 are not strong enough for a strong paper. Most of the issues mentioned under cascading effects should be quantified and if possible estimated econometrically. For instance, what proportion of food crops was wasted? What is the level of increase in transport cost due to floods and droughts – 50 or 70%?
Author Response
Thank you for your comments.
Find attached detailed of how the comments were addressed.

Reviewer 2 Report
The article aims to investigate the impact climate change has on agricultural infrastructure, using two communities in Nigeria as cases. The data consist of interviews with eight key informants. The authors conclude that climate change has had a negative impact on agricultural infrastructure.
The topic is important, but sadly the quality of the article does not meet academic standards. First, it lacks a proper theoretical and methodological discussion. There is a literature that deals with the impact on droughts and floods in rural communities in Africa that is not cited in the text. The method section consist of a very short introduction to the two communities without any further information on why these were chosen. In addition, the author do not explain how the key informants were identified and samples, how the interview were conducted. They also fail to present the findings from the interviews in a proper way.
More important though is that one cannot focus on one single event to discuss the impact of climate change. To analyse its impact you need to conduct a longitudinal study that analyses the impact of volatility in weather conditions over time. Lastly, to me it makes limited sense why the authors rely only on interviews with key informants. Why not complement it with impact reports, surveys etc. And who not interview farmers.
Author Response
Thank you for your comments.
Find attached details of how the comments were addressed.

Round 2
Reviewer 2 Report
My main concerns with the previous draft was the lack of a proper methodological and theoretical discussion, the focus on a single event and the lack of additional sources apart from the key informants.
The theoretical part has been improved, which makes the line of reasoning more accessible. However, I still lack a section which elaborate how theory will be used in this study.
The method part has also been improved and is to my satisfaction.
My main concern is that the paper does still not really deal with the effects of climate related events, at least not in the strict sense of the term. There is a lot of useful information on how different actors perceive the effects, but to follow the aim strictly this must be combined with data on infrastructure damage. If such data do not exist I recommend the author(s) to re-phrase the titel and the introduction to make it clear to the reader that the paper is about perceptions, which could vary for a number of reasons that may not have less to do with the climate related events.
Author Response
2 main concerns raised by the reviewer are:
The lack of a section to elaborate how theory will be used in this study.
This is now addressed on page 6 by including a theoretical framework of the study.
The need to combine data on infrastructure damage to understand the effect of climate related events, and to re-phase the title of the paper.
This is addressed in page 12 section 5.3
Analysis of the level of climate impacts on agrarian infrastructure systems is included. Because this is also based on respondents perceptions, the researcher has re-phrased the title, abstract and introduction. The title now reads "Perceived Impacts of Climate Related Events on Agrarian Infrastructures and the Associated Cascading Effects on Livelihood Systems in Plateau State, Nigeria".
